# Automated Exoplanet Transit Detection from Stellar Light Curves*

**Tommaso Dall'Amico**[†]
School of Aerospace Engineering
Tsinghua University
Beijing, CN
cyc25@mails.tsinghua.edu.cn

**Mikhail Kuvakin**[‡]
Department of Computer Science
Tsinghua University
Beijing, CN
mkekwj25@mails.tsinghua.edu.cn

**Lamat Foschini Atichi**[§]
School of Economics & Management
Tsinghua University
Beijing, CN
Lamat.foschini@student.unisg.ch

## 1 Introduction

The discovery of exoplanets is a crucial frontier in modern astrophysics. It expands our understanding of planetary formation, contextualizes the uniqueness of our own solar system, and drives the search for potentially habitable worlds. Just as ancient explorers like Herodotus gathered knowledge about islands of habitability outside the known world, modern astronomers seek out habitable zones beyond our local cosmic neighborhood. Until recent years, the field relied heavily on classical approaches and manual interpretation of potential exoplanet candidates. These methods are labor-intensive, subject to human error, and often struggle to isolate weak signals hidden in noise. The recent introduction of Machine Learning has driven significant progress, leading to the automated discovery of many new potential exoplanets. In this course project, we aim to contribute to this intersection of Planetary Science and ML by bringing state-of-the-art Time-Series Intelligence techniques into the existing framework of exoplanet discovery. Specifically, we plan to focus on the automated search for smaller exoplanets with weak or non-periodic transit signals, which are currently difficult for existing architectures to recognize.

## 2 Background

There are various methods to detect exoplanets:

- **Direct imaging:** Relies on isolating and measuring the actual light emitted or reflected by the exoplanet itself.

- **Microlensing:** Relies on the chance alignment of two stars with an observer. As one star crosses behind the other, its gravity bends the light, causing a brightness spike. If a planet is present, its gravity causes an additional detectable anomaly.

- **Astrometry:** Involves tracking the minute, physical wobbles of a star caused by the gravitational pull of orbiting planets.

---

*Course project for Time Series Intelligence class, Spring 2026
[†]2025280176
[‡]2025280345
[§]2026400111

However, the most fundamental and effective method is **Transit photometry**. This method detects a dip in starlight: as an exoplanet passes in front of its host star, it blocks a portion of the light, reducing the flux of photons reaching the telescope. By recording time series data of this light intensity, we can search for periodic "dips" that indicate the presence of a planetary system.

Crucially, a dimming star does not necessarily guarantee the presence of a planet. A variety of false positives can mimic transit signals:

- **Eclipsing binaries:** Two stars orbiting one another that periodically block each other's light.
- **Background eclipsing binaries:** Distant eclipsing star systems whose light blends with the target star in the telescope's large pixels.
- **Hierarchical systems:** Complex, multi-star systems where internal eclipses mimic a planetary transit.
- **Stellar variability:** Intrinsic, natural fluctuations in a star's brightness over time.
- **Instrumental or systematic noise:** Artificial signals introduced by telescope hardware or data processing pipelines.

To distinguish genuine exoplanets from these phenomena, astronomers rely on a variety of derived metrics and diagnostic tests.

## 2.1   Basics of Light Curve Analysis

In transit photometry, the observed flux can be modeled as:

$$F_{\text{obs}}(t) = F_\star(t) + s(t) + n(t),$$

where $F_\star(t)$ represents the intrinsic stellar and instrumental baseline, $s(t)$ is the transit signal, and $n(t)$ is the noise. Since a transit is usually a short-duration, low-amplitude dip, while unwanted trends evolve over longer timescales, the data must first be detrended. A smooth baseline $T(t)$ is estimated and removed:

$$F_{\text{det}}(t) = F_{\text{obs}}(t) - T(t) \quad \text{or} \quad F_{\text{det}}(t) = \frac{F_{\text{obs}}(t)}{T(t)}.$$

The goal is to suppress slow variability while preserving the short-timescale transit morphology.

A periodic signal is characterized by an orbital period $P$, epoch $t_0$, depth $\delta$, and duration $\tau$. For truly periodic signals, transit times occur approximately at:

$$t_k \approx t_0 + kP, \quad k \in \mathbb{Z}$$

Periodicity is vital because several weak events can be combined coherently to increase detectability. For a trial period $P$, the phase is defined as:

$$\phi(t) = \left\{ \frac{t - t_0}{P} \right\}, \quad \text{where } \{a\} = a - \lfloor a \rfloor,$$

mapping all data points into the interval $[0, 1)$. Plotting flux as a function of $\phi$ creates a folded light curve. Here, repeated transits align in phase, random fluctuations average out, and the transit shape is isolated.

Using phase folding, we can compute a periodogram—a detection score evaluated across various trial periods. A common method is the Box Least Squares (BLS) periodogram. BLS folds the light curve at a trial period and fits a simplified box-shaped transit model to the data. The model assumes constant stellar flux outside the transit and a fixed decrease during the transit:

$$f(\phi) = \begin{cases} 1 - \delta, & \text{if } \phi \in [\phi_{\text{in}}, \ \phi_{\text{in}} + \Delta\phi] \\ 1, & \text{otherwise} \end{cases}$$

where $\phi$ is the phase, $\delta$ is the transit depth, and $\Delta\phi$ is the transit duration. If the trial period aligns with the true orbital period, the box provides a significantly better fit than a flat model. The BLS process evaluates trial periods iteratively to construct a function $\mathcal{P}(P)$ that identifies periods where the data appears most transit-like.

## 2.2 Metrics

**Signal-to-Noise Ratio (SNR):** The most generic measure of detection significance. It compares the strength of the candidate signal to the characteristic background noise level. A higher SNR indicates the signal stands out clearly above background fluctuations.

**Signal Detection Efficiency (SDE):** Derived from the periodogram, SDE measures how exceptional a peak is relative to the overall distribution of peaks in the search spectrum. It acts as a normalized peak height:

$$\text{SDE} \sim \frac{\mathcal{P}(P) - \text{baseline}}{\text{scatter of the spectrum}}$$

A high SDE means the trial period is unusually prominent compared to the background structure of the search.

**Single Event Statistic (SES):** Measures the significance of an individual transit-like dip in isolation. A high SES alone is insufficient to confirm a planet, as it may just represent one strong, non-repeating event.

**Multiple Event Statistic (MES):** Evaluates the stacked significance of a repeated signal after aligning multiple events at a trial period. MES rewards repeatability and consistent timing. A high SES coupled with a low MES suggests inconsistent repetition.

These quantities complement one another: SNR provides overall detectability, SES measures individual event strength, MES verifies coherent repetition, and SDE confirms the candidate period's prominence in the search spectrum.

## 2.3 Diagnostic tests

**Centroid analysis:** Tracks the apparent position of the center of light within the photometric aperture over time. If a transit occurs on the target star, the centroid remains stable. If the dip is caused by a nearby contaminating source, the centroid will shift. The centroid is calculated as the flux-weighted average position of pixels $(x_i, y_i)$ with fluxes $f_i$:

$$x_c = \frac{\sum_i x_i f_i}{\sum_i f_i}, \quad y_c = \frac{\sum_i y_i f_i}{\sum_i f_i}$$

**Difference image:** Constructed by comparing in-transit and out-of-transit pixel images to measure localized flux changes. This helps determine exactly where on the pixel map the light is decreasing, distinguishing genuine on-target transits from background false positives.

**Background flux and contamination:** Because TESS pixels are large, apertures often include flux from nearby stars. This can dilute transit depths or allow background eclipsing binaries to mimic shallow planetary transits. Machine learning models must process background time series to distinguish these astrophysical phenomena.

**Odd/even transit test:** Checks for eclipsing binaries by comparing the depths of alternating transits. If the search pipeline mistakenly identifies a period at half the true orbital period, primary and secondary eclipses fold together. Unequal depths between odd and even events strongly indicate a binary star system rather than a planet.

## 2.4 Basic pipeline

Raw data the Kepler/TESS Missions is processed by the Science Processing Operations Center (SPOC). SPOC generates calibrated pixels, light curves, and centroids, and performs initial transit searches to identify Threshold Crossing Events.

Working with SPOC preprocessed data, automated pipelines typically follow three stages:

1. **Search:** Identifying transit-like signals from the light curves.
2. **Vetting:** Classifying candidates as likely planets or false positives.
3. **Validation:** Confirming a vetted candidate is statistically guaranteed to be a real planet.

# 3 Related works

The majority of current literature focuses on the vetting and validation stages. For instance, *DART-Vetter* acts as a triage model using a highly compact preprocessing pipeline, relying almost entirely on folded and binned photometry to separate candidates from astrophysical false positives [Fiscale et al., 2025]. Conversely, *ExoMiner* and its next generation *ExoMiner++ 2.0* utilize a much richer set of SPOC-derived representations, incorporating multiple folded flux views, centroid motion, difference images, and scalar diagnostics to vet Threshold Crossing Events[Valizadegan et al., 2022, Martinho et al., 2026]. Models like *RAVEN* combine classical detrending and BLS-based detection with tailored feature engineering for probabilistic validation[Hadjigeorghiou et al., 2025].

Fewer studies have attempted to automate the initial search phase. Notably, work [Salinas et al., 2025] bypasses phase-folding entirely, operating directly on full, standardized TESS light curves augmented with centroid and background time series. However, current architectures, mostly utilizing CNNs or standard Transformers, still struggle to accurately identify small planets.

# 4 Challenges

A primary challenge is detecting small exoplanets, whose minimal influence on stellar light makes them easily confused with instrumental noise and stellar variability. Additionally, the automated period search is prone to harmonic aliases. For example, the presence of identical primary and secondary eclipses in a binary system can cause an algorithm to mistakenly report a period of $P/2$. Conversely, sparse or irregular sampling can make incorrect repetition patterns appear plausible, leading to spurious peaks at $2P$ or other harmonics. Because the initial search stage can report incorrect periods, subsequent vetting stages are strictly necessary to evaluate the folded light curve, centroid behavior, and odd/even differences to ensure the signal is physically consistent with a genuine planet.

# 5 Objectives and proposed methodology

We aim to enhance the automated search for transit-like signals using state-of-the-art Time Series Intelligence techniques. By improving algorithmic sensitivity, we intend to detect smaller, one-transiting, or non-periodic exoplanets that produce weak light curves typically missed by current models. Following detection, we will process these candidates through established vetting models to compile a database of newly validated planet candidates.

To achieve this, we will leverage the following strategies:

- **Advanced Architectures:** Implementing newer sequence modeling paradigms, such as a Mamba and Attention hybrid architecture, specialized for the nuances of astronomical time series.

- **Cross-Domain Training:** Utilizing both Kepler and TESS data streams to double-check observations, reduce instrumental noise bias, and improve generalization.

- **Data Augmentation:** Expanding the training set by generating artificial data and injecting false labels to mitigate class imbalance and improve the model's robustness against false positives.

# 6 Dataset

The rawest data we will utilize comes in two formats:

- **Target Pixel Files (TPF):** Target-level data used when we have a specific star and require maximum control over aperture selection, background treatment, and custom photometry as shown in figure...

- **Full Frame Images (FFI):** Field-level data used to scan large populations, including stars absent from original target lists as in Figure ...

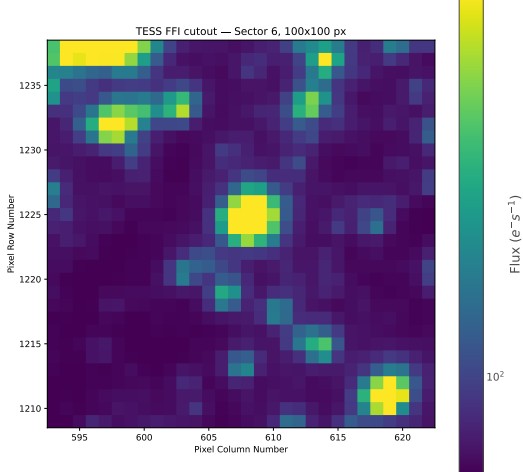

Figure 1: TESS FFI data example

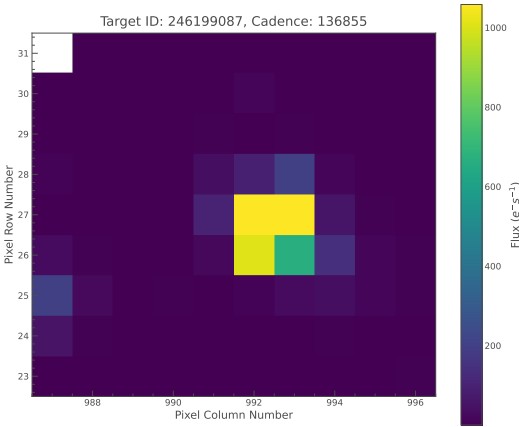

Figure 2: Kepler TPF data example

Data will be sourced from the Kepler and TESS Science Processing Operations Center (SPOC) archives hosted at MAST (https://archive.stsci.edu/). We will interface with this data using the `lightkurve` Python library (https://pypi.org/project/lightkurve/).

# 7   Current progress and project schedule

We have completed a comprehensive literature review on transit photometry and established a strong understanding of current pipeline methodologies, identifying clear opportunities for the application of novel Time Series Intelligence architectures. The schedule for the future work is described in Table 1.

Table 1: Project Schedule

| Time Period | Primary Tasks | Implementation Details |
| --- | --- | --- |
| Week 10 | Proposal Submission | Finalize the written project proposal and prepare the preliminary presentation for initial review. |
| Week 11–12 | Data Acquisition | Access the MAST archive, analyze data structure, assess the computational complexity of candidate algorithms, and generate additional synthetic transit data for training. |
| Week 13–14 | Pipeline Development | Implement the core search algorithm and extract pre-computed transit parameters through high-throughput processing of light curve data. |
| Week 15 | Vetting and Validation | Deploy existing classification architectures to filter astrophysical false positives and obtain high-probability exoplanet candidates. |
| Week 16 | Final Presentation | Summarize research findings, generate performance visualizations, and compose the final project presentation. |

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
