# OpenReview forum: "Automated Exoplanet Transit Detection from Stellar Light Curves"
_tsinghua.edu.cn/THU/2026/Spring/ANM — THU 2026 Spring ANM Submission_

### Official Review · Reviewer_vwPw · 2026-05-12

**Rating:** 7
**Confidence:** 4

**Summary:**

The proposal aims to improve automated exoplanet detection using modern time-series architectures (e.g., Mamba + Attention hybrid), focusing on small or weakly periodic transit signals missed by current methods. The authors plan to use Kepler and TESS data, apply data augmentation, and integrate existing vetting pipelines to produce a validated candidate list. The work is positioned between the search and vetting stages, with a clear nod to domain challenges like harmonic aliases and false positives.

**Strengths:**

1. Well-motivated domain background – The proposal shows a solid understanding of transit photometry, noise sources, and diagnostic metrics (SNR, SDE, SES, MES), which is rare and valuable for a non-astrophysics team.

2. Clear problem targeting – Focus on small, non-periodic, or weak signals is appropriately ambitious for a course project and addresses a real gap.

3. Reasonable technical plan – Mention of Mamba + Attention, cross-domain training, and synthetic augmentation shows awareness of current time-series SOTA beyond standard CNNs/Transformers.

**Weaknesses:**

1. Missing baseline and evaluation metric specification – No explicit mention of which existing model(s) will be compared against (e.g., BLS, ExoMiner, or Salinas’ Transformer). Without this, “improvement” is hard to measure.

2. Vague on “non-periodic” detection – The proposal repeatedly mentions weak or non-periodic transits, but transit photometry fundamentally relies on periodicity for stacking (MES). How the hybrid architecture handles truly aperiodic signals is not explained.

3. Over-reliance on synthetic data – Generating synthetic transits is useful, but validation will require real injected signals or labeled false positives. The proposal does not describe how to avoid overfitting to synthetic patterns.

**Questions:**

1. How will you quantitatively compare your method against BLS or a simple Transformer baseline on weak signals? What is the primary metric (e.g., MES recovery rate, SDE at low SNR, AUC for injection tests)?

2. For “non-periodic” transits: without periodicity, how do you avoid conflating single-event noise with real transits? Are you targeting non-strictly-periodic (e.g., transit timing variations) or truly aperiodic events?

3. Which existing vetting model(s) will you use (DART-Vetter, ExoMiner, RAVEN, or a simplified version)? Will you re-train or run them as black-box classifiers?

4. What is your plan for class imbalance (genuine planets are rare)? Synthetic injection ratio and train/test split on real data?

---

### Official Review · Reviewer_yMfX · 2026-05-17

**Rating:** 7
**Confidence:** 4

**Summary:**

This proposal is a framework aiming to improve the detection of exoplanets using modern time-series intelligence techniques. The work will mainly focus on the detection of small exoplanets, which weak signals make them unaccurately treated by current methods. The authors will intend to automate transit photometry using Mamba and Attention hybrid architecture, before testing their model on the Kepler and TESS datastreams to produce a candidate list of exoplanets.

**Strengths:**

-Very original and ambitious subject, which motivations and state of the art have been well documented.

- For such a novel and less explored topic, restricting the scope of the study to improving the current baselines (by improving the detection of some specific exoplanets) rather than buildling from scratch a whole new framework is a very good idea.

- Risks and limitations have also been clearly identified, and will surely be useful for the sake of false-positive detection

**Weaknesses:**

- The pipeline part could have been more expanded. In particular, the overall neural network architecture is missing, and explaining why you are using Mamba/Attention would have been interesting.

- Evaluation metrics and methodology are also lacking in the proposal

- Overall, almost half of the paper has been dedicated to the mathematical and physical detection of exoplanets. Without erasing how interesting it is, I think it would have been benefic to shorten this technical part, and instead develop more the time series part on which more people can understand and give feedback.

**Questions:**

Why are you using data coming from two different sources? Are they complementary? Do they come from the same mission?

---

### Official Review · Reviewer_qtXs · 2026-05-17

**Rating:** 8
**Confidence:** 3

**Summary:**

This proposal aims to improve automated exoplanet detection by applying state-of-the-art time-series architectures for transit-like signals. The project focuses on small planets with weak or non-periodic transit signals, which are currently poorly handled by CNNs and standard Transformers. The team plans to use public Kepler and TESS data via the lightkurve library, apply cross-domain training across both sources, and augment training data with synthetic data to address class imbalance. Candidates will be passed through established vetting models to compile a database of newly validated planet candidates.

**Strengths:**

- The background section is thorough, which demonstrates domain expertise and sets a strong foundation for the technical work.
- Focusing on the search stage rather than vetting is a well-justified choice. The related work section observes that most current literature addresses vetting and validation, making the search stage a genuine gap.
- The related work is up-to-date and specific, citing recent and relevant papers.
- Using publicly available, well-documented data (Kepler/TESS via MAST and lightkurve) ensures reproducibility and removes data access as a project risk.
- The plan to apply cross-domain training across Kepler and TESS is a reasonable strategy for improving generalisation and reducing mission-specific instrumental bias.

**Weaknesses:**

- The methodology section is underspecified. Not specified how Mamba and Attention hybrid are combined, what the input representation is, and how the model produces a detection score.
- No evaluation metrics or baselines are mentioned.
- The data augmentation strategy is mentioned but not described.

**Questions:**

- What exactly is the input to the Mamba-Attention model? How is the model's output structured?4
- What is the planned comparison baseline? How will you isolate the contribution of the Mamba component specifically?

---

### Official Review · Reviewer_Syd7 · 2026-05-17

**Rating:** 8
**Confidence:** 4

**Summary:**

This proposal outlines a project to automate the detection of exoplanet transits from stellar light curves using Time-Series Intelligence techniques. The authors specifically target smaller exoplanets with weak or non-periodic signals, which are notoriously difficult to detect using existing architectures. The methodology proposes using a hybrid Mamba and Attention architecture, cross-domain training on Kepler and TESS data, and data augmentation with synthetic data. The data will be sourced from the MAST archives utilizing Target Pixel Files (TPF) and Full Frame Images (FFI).

**Strengths:**

The proposal provides an excellent, comprehensive background on transit photometry, including a clear explanation of false positives like eclipsing binaries and stellar variability.
The metrics (SNR, SDE, SES, MES) and diagnostic tests (centroid analysis, difference image) are well-defined and demonstrate a deep understanding of the astronomical domain.
The proposed methodology introduces a modern approach by suggesting a Mamba-Attention hybrid, which is highly relevant for capturing the nuances of astronomical time series.
The inclusion of cross-domain training using both Kepler and TESS datasets is a strong strategy to reduce instrumental bias and improve generalization.

**Weaknesses:**

While the proposal mentions using a "Mamba and Attention hybrid architecture," it lacks technical details on exactly how these two distinct sequence modeling paradigms will be integrated.

**Questions:**

How specifically do you plan to combine the Mamba and Attention modules within your proposed architecture to handle these specific light curves?

Regarding data augmentation, what specific generative techniques will be used to create realistic "artificial data" and inject false labels without introducing unrealistic artifacts into the training set?

---

### Official Review · Reviewer_ivLW · 2026-05-18

**Rating:** 8
**Confidence:** 3

**Summary:**

The project aims to improve automated detection of exoplanets (particularly small, weakly transiting, or non-periodic candidates) from stellar light curves using advanced time-series machine learning. The authors propose moving beyond current CNN/Transformer models by implementing hybrid architectures (Mamba + Attention) and leveraging cross-domain training (Kepler + TESS data) and data augmentation

**Strengths:**

1. addresses a genuine gap: existing ML methods struggle with small or irregular transit signals
2. well-structured background on transit photometry, key metrics (SNR, SDE, MES), false positives (eclipsing binaries, stellar variability), and diagnostic tests (centroid analysis, odd/even transit test)
3. references state-of-the-art models (ExoMiner, DART Vetter, RAVEN) and notes that few efforts focus on the search phase (vs. vetting/validation)

**Weaknesses:**

The proposal describes what each metric means (e.g., high SDE indicates a prominent period), but it does not state how those metrics will be used to quantitatively assess the model's performance. Missing elements include: a baseline method to compare against (e.g., classical BLS or ExoMiner), a labeled test set of confirmed planets and false positives, success thresholds (e.g., “achieve 15% higher recall for small planets at a false positive rate below 5%”), and a clear decision rule for whether a detection is correct

---

### Official Review · Reviewer_SmDj · 2026-05-18

**Rating:** 8
**Confidence:** 3

**Summary:**

This proposal explores the opportunity for automated exoplanet discovery via astronomical time-series data analysis, with the focus on enhancing the sensitivity of existing pipelines for detection of weak or non-periodic transiting exoplanets. The approach leverages state-of-the-art architectures, including Mamba and Attention hybrid, and data augmentation techniques on Kepler and TESS datasets.

**Strengths:**

- Ambitious topic choice with potential scientific significance, the proposal reflects investment and collective efforts of the team to understand the problem matter closely;
- Demonstrated understanding of the existing approaches and their shortcomings, and integration of the state-of-the-art time-series modelling techniques;
- A straightforward application of time-series analysis for addressing an observed critical gap in exoplanet-related signal processing.

**Weaknesses:**

- At the current stage of the project proposal, the methodology is described conceptually, a concrete step-by-step pipeline was not fully defined. Similarly, evaluation, metrics, expected results are not yet specified;
- Despite authors' demonstrated grasp on the topic and careful literature review, further execution and validation may require additional specialized astronomical domain knowledge.

**Questions:**

- How robust do you anticipate the model to be to sensor errors, noise or anomalies in the light curves?
- Have computational requirements for model training/validation/data augmentation/testing been estimated, is the proposed attempt achievable given available computational resources?

---

### Official Review · Reviewer_FbqQ · 2026-05-18

**Rating:** 5
**Confidence:** 4

**Summary:**

[AI Review] This proposal titled "Automated Exoplanet Transit Detection from Stellar Light Curves" aims to use a Mamba + Attention architecture for exoplanet transit detection but fails to deliver substantive technical content. The paper reads as a literature review with project aspirations rather than a concrete technical proposal, oscillating between search and vetting tasks without committing to either. 50% of the content is textbook background that should be compressed, while core components like architecture definition, experimental design, and methodology remain unspecified. With significant revision including architecture diagrams, task definition, and experiment plans, the paper could reach a score of approximately 7/10.

**Strengths:**

1. Addresses a relevant and scientifically meaningful problem in automated exoplanet detection from stellar light curves.
2. Identifies the potential of modern sequence modeling approaches (Mamba + Attention) for time-domain astronomy.
3. Acknowledges the closest prior work (Salinas et al. 2025), showing awareness of recent literature.
4. Considers cross-domain applicability between Kepler and TESS missions, showing ambition for generalizability.
5. Proposes data augmentation via false label injection, indicating some thought about training challenges.

**Weaknesses:**

1. No architecture defined despite mentioning "Mamba + Attention" — no diagram, layer specifications, or input/output mappings provided (severity 10).
2. Fundamental task ambiguity: oscillates between search (finding candidates) and vetting (validating candidates) without clearly defining which problem is being solved (severity 10).
3. Zero experimental design — no metrics, baselines, dataset splits, or success criteria specified (severity 9).
4. Undefined cross-domain training approach: proposes Kepler↔TESS mixing without acknowledging or addressing the significant domain gap between instruments (severity 9).
5. Vague data augmentation: "injecting false labels" uses undefined terminology and lacks methodological detail (severity 9).
6. Only 6 references, missing seminal works including Shallue & Vanderburg 2018 and Kovács et al. 2002 (BLS).
7. 50% of paper is textbook background (transit photometry, BLS) that should be compressed to 25%.
8. No differentiation from closest prior work (Salinas et al. 2025) despite citing it.
9. Timeline allocates only 2 weeks for core algorithm development, which is unrealistic.
10. Non-periodic detection claim in introduction is unsupported by any proposed method.
11. Incomplete proofreading: text contains "figure..." with ellipsis indicating unfinished sections.

**Questions:**

1. Can you provide a concrete architecture diagram with layer specifications, dimensions, and input/output mappings for the proposed Mamba + Attention model?
2. Is this project addressing the search problem (identifying transit signals in raw light curves) or the vetting problem (classifying candidates as true planets vs false positives)? Please commit to one clearly.
3. What specific metrics (precision, recall, F1, AUC-ROC, etc.) will you use to evaluate performance, and what are your success criteria?
4. How do you plan to handle the domain gap between Kepler and TESS data (different cadences, bandpasses, noise properties)? Will you train separate models, use domain adaptation, or fine-tune?
5. What exactly does "injecting false labels" mean for data augmentation? Are you synthesizing transit signals, injecting false positives, or something else?
6. How does your proposed approach differ from or improve upon Salinas et al. 2025, which you cite as the closest prior work?
7. Given the 2-week timeline for core algorithm development, what is the minimum viable experiment you plan to demonstrate?

---

### Official Review · Reviewer_6vSx · 2026-05-18

**Rating:** 7
**Confidence:** 2

**Summary:**

The authors propose a hybrid Mamba-Attention architecture, augmented by cross-domain training (using Kepler and TESS data) and synthetic data generation. The proposal demonstrates an exceptionally strong grasp of astronomical domain knowledge and clearly identifies a valid gap in the literature. However the extensive astronomical background leaves less room for concrete machine learning methodology.

**Strengths:**

1.Exceptional Domain Knowledge: The proposal provides a highly comprehensive and accurate summary of transit photometry, including fundamental equations, periodogram analysis (BLS), key metrics (SNR, SDE, SES, MES), and diagnostic tests.
2.Astute Problem Positioning: The authors correctly identify that while much recent ML literature focuses on the downstream "vetting" and "validation" stages (e.g., ExoMiner, DART-Vetter), the upstream "search" stage remains heavily reliant on classical methods and struggles with small planets.
3.Sound Technical Direction: Proposing a Mamba-Attention hybrid is a highly relevant choice for this specific domain. Mamba's state-space modeling is well-suited for the extremely long sequence lengths of high-cadence stellar light curves, while Attention can capture specific localized transit features

**Weaknesses:**

1.Imbalanced Content: Out of 6 pages, approximately 3.5 pages are dedicated to standard astronomical background, while the actual proposed machine learning methodology is condensed into a single bulleted list of just a few sentences.
2.Undefined ML Task Formulation: It is entirely unclear how the model will operate on the data. Is this formulated as a sequence-to-sequence anomaly detection task, a segmentation task (labeling in-transit vs. out-of-transit timesteps), or a classification task on folded phase curves?

**Questions:**

1.How exactly is the machine learning task formulated? What are the specific inputs to the model, and what does the final output layer predict?
2.Can you provide a concrete architectural diagram or description detailing how the Mamba and Attention modules are integrated?
3.How will you preprocess the data to overcome the differing sampling rates and instrumental noise profiles between the Kepler and TESS datasets for your cross-domain training?
4.What quantitative metrics will you use to evaluate your model, and what baseline models will you compare your results against?